# High-Resolution Microscopic Characterization of Tunneling Nanotubes in Living U87 MG and LN229 Glioblastoma Cells

**DOI:** 10.3390/cells13050464

**Published:** 2024-03-06

**Authors:** Nicole Matejka, Asieh Amarlou, Jessica Neubauer, Sarah Rudigkeit, Judith Reindl

**Affiliations:** Institute for Applied Physics and Measurement Technology, University of the Bundeswehr Munich, 85577 Neubiberg, Germany; asieh.amarlou@tum.de (A.A.); jessica.neubauer@unibw.de (J.N.); sarah.rudigkeit@unibw.de (S.R.); judith.reindl@unibw.de (J.R.)

**Keywords:** tunneling nanotube, glioblastoma, cellular communication, cancer treatment, live-cell microscopy, high-resolution microscopy

## Abstract

Tunneling nanotubes (TNTs) are fine, nanometer-sized membrane connections between distant cells that provide an efficient communication tool for cellular organization. TNTs are thought to play a critical role in cellular behavior, particularly in cancer cells. The treatment of aggressive cancers such as glioblastoma remains challenging due to their high potential for developing therapy resistance, high infiltration rates, uncontrolled cell growth, and other aggressive features. A better understanding of the cellular organization via cellular communication through TNTs could help to find new therapeutic approaches. In this study, we investigate the properties of TNTs in two glioblastoma cell lines, U87 MG and LN229, including measurements of their diameter by high-resolution live-cell stimulated emission depletion (STED) microscopy and an analysis of their length, morphology, lifetime, and formation by live-cell confocal microscopy. In addition, we discuss how these fine compounds can ideally be studied microscopically. In particular, we show which membrane-labeling method is suitable for studying TNTs in glioblastoma cells and demonstrate that live-cell studies should be preferred to explore the role of TNTs in cellular behavior. Our observations on TNT formation in glioblastoma cells suggest that TNTs could be involved in cell migration and serve as guidance.

## 1. Introduction

The ability of cells to rapidly adapt and reorganize themselves when their environment and living conditions change has been an important aspect of global research interest since the beginning of cell research. Efficient cellular communication is a key driver in wound healing, immune response, embryonic development, and more, as well as in disease control and prevention. One of medicine’s greatest challenges is the fight against cancer. According to the International Agency for Research on Cancer’s (IARC) World Cancer Report 2020, 29.8% of premature deaths from noncommunicable diseases (NCDs) worldwide in 2016 were due to cancer, making it the leading cause of death worldwide [1]. Therefore, the role of cellular communication in cancer treatment has been intensively studied to better understand the complexities involved in the interaction of different cell types in the tumor microenvironment, cancer invasion, and progression, and to identify novel therapeutic strategies [2,3,4,5,6].

Cells can communicate with each other through a variety of mechanisms: exosomes, gap junctions, extracellular vesicles, the secretion of molecules, and more. However, a very fast and efficient way to communicate is the interconnection of cells by tunneling nanotubes (TNTs). This type of intercellular communication was discovered in rat pheochromocytoma PC12 cells by Rustom et al. [7] using 3D live-cell microscopy in 2004. These “nanotubular highways” are membrane tunnels with a diameter from 50 nm to 1500 nm that directly connect cells over long distances of more than 100 µm, providing optimal communication between them [7,8]. Through these TNTs, cells can form a large communication network with a high connection rate, allowing for a high level of functional exchange [9]. It has been shown that TNTs can transport cargoes such as mitochondria, calcium signals, death signals, p-glycoprotein, microRNAs, and nanoparticles [10,11,12,13,14]. In recent years, TNTs have been found in many other cells, both cancerous and non-cancerous cell lines and tissues [8,15,16,17,18,19,20,21]. It has been found that there is a great diversity in the morphology, composition, and function of TNTs [22]. Also, the formation of TNTs differs among cell types. It has been shown that TNTs are dominantly formed in neuronal PC12 cells by actin-driven filopodia growth [7]. In contrast, TNTs are formed in immune cells such as natural killer cells, macrophages, or T cells, as well as in normal rat kidney cells by cell separation due to cell movement [8,23]. It is also possible that both formation mechanisms can occur in one cell line, as has been shown for the human embryonic kidney (HEK) 293 cell line [23,24]. The exact process of TNT formation is still unclear, and there are many theories about the exact interplay of molecular components that may play a role in TNT formation [25]. TNTs usually appear as straight, thin membrane connections between distant cells, which have a very flexible shape because they dynamically shrink or expand as the cells move. The length can be regulated in a range from a few microns to over 100 µm until the distance between the connected cells becomes too large and the TNT disappears. However, the role of TNTs in cellular communication, especially in cancer cells, is still unknown [8]. Nevertheless, due to their strong ability to form complex and functional communication networks between different cells, TNTs seem to be a promising target for new cancer therapy strategies [8,21,26,27,28]. For a novel therapy, TNTs could either be inhibited to disrupt the functional communication network or used as a drug delivery system. Several studies have shown that nanoparticles can be transferred to other cells via TNTs [17,29,30,31] and, therefore, the possibility of exploiting the TNT network in nanomedicine has been strongly discussed [32,33]. Especially for very aggressive cancers such as glioblastoma, TNTs could provide a much-desired opportunity to improve its poor prognosis. 

Glioblastoma multiforme is the most common malignant human brain tumor. Despite multimodal treatment combining surgery, concurrent and/or adjuvant chemotherapy, and radiotherapy, the tumor often recurs within 1–2 cm of the primary tumor [34]. The median survival of glioblastoma patients is less than 15 months and the five-year survival rate is less than 10% [35,36,37]. Due to its aggressive nature, consisting of high infiltration rates, uncontrolled cell growth, intra-tumoral heterogeneity, genomic instability, and its strong ability to develop high resistance to chemotherapy and radiotherapy, the treatment of glioblastoma multiforme results in the aforementioned poor prognosis [4,38,39]. It has been shown that TNT-like membrane structures are present in brain tumors and form functional, resistant communication networks [40]. Furthermore, communication via TNTs or TNT-like structures has been reported several times in glioblastoma cell lines [41,42,43,44,45,46,47,48,49]. These findings make TNTs a potential target for improving the treatment of glioblastoma [8,18,23,50,51,52,53]. However, knowledge is still very limited. Thus, the need for further research on TNTs to achieve the goal of developing new therapeutic approaches is high [8,23]. 

In this study, we characterize TNTs in the glioblastoma cell lines U87 MG and LN229 in detail by using high-resolution live-cell microscopy and share our knowledge of the microscopy methods that enable TNT research in living cells. In particular, we found out which membrane-labeling method works best for TNT research using live-cell confocal and stimulated emission depletion (STED) microscopy. We accurately measured the diameter of the TNTs using the method of van Steensel et al. [54] after obtaining the TNTs by live-cell STED microscopy. We investigated how TNTs are formed in these glioblastoma cell lines and how the TNTs behave during cell movement. We examined the length and lifetime of the TNTs. Furthermore, we summarized the morphological features of the TNTs and analyzed at which point during cell fixation that the TNTs most frequently break. We show why the study of sensitive TNTs should preferably be performed by live-cell microscopy. With our study, we provide new insights into the fundamental properties of TNTs in glioblastoma cells and give hands-on guidance for the challenging microscopy of small membrane structures to advance TNT research.

## 2. Materials and Methods

### 2.1. Cell Culture, Labeling, and Fixation

The human glioblastoma cell lines U87 MG (ATCC, Manassas, VA, USA, HTB-14) and LN229 (ATCC, Manassas, VA, USA, CRL-2611) were kindly provided by the Institute of Radiation Medicine (Helmholtz Zentrum München GmbH, 85764 Neuherberg, Germany) and the Department of Radiation Oncology (Ludwig-Maximilian-University, 80539 München, Germany), respectively. The cells were cultured in DMEM, high-glucose medium (Merck KGaA, Darmstadt, Germany, D6429) supplemented with 10% FCS (Merck KGaA, Darmstadt, Germany, F7524), and 1% penicillin/streptavidin (Merck KGaA, Darmstadt, Germany, P4333) and maintained at 37 °C in a controlled atmosphere with 5% CO_2_ and 95% humidity. For 3D culturing, the U87 cells were embedded in Matrigel (Corning, New York, NY, USA, 356231) at a concentration of 1,000,000 cells/mL. This Matrigel is phenol-red-free and, therefore, more suitable for high-resolution microscopy.

The cells were seeded on the day before the experiment in glass-bottom dishes, either a µ-Dish 35 mm (Ibidi GmbH, Gräfeling, Germany, 81158) or a CELLview cell culture dish 35 × 10 mm with 4 compartments (Greiner Bio-One GmbH, Frickenhausen, Germany, 627870). A seeding density of 65,000 cells/mL was chosen for the Ibidi dishes (2 mL total volume; 37,143 cells/cm^2^) and 55,000 cells/mL for the Greiner dishes (1 mL volume per well; 42,308 cells/cm^2^). For the microscopy of the 3D cell culture, the Matrigel-embedded cells were seeded on a µ-Slide 15 Well 3D Glass Bottom (formerly µ-Slide Angiogenesis Glass Bottom) (Ibidi GmbH, Gräfeling, Germany, 81507) at 10,000 cells per well (10 µL volume per well; 43,478 cells/cm^2^). After 30 min of incubation time in the cell incubator to allow the Matrigel to gel, 50 µL of growth medium was added per well. All glass-bottom products used had a glass thickness of (170 ± 15) µm and were, therefore, suitable for high-resolution microscopy.

The following stains were used for the plasma membrane labeling: CellMask™ Plasma Membrane Stains (Thermo Fisher Scientific Inc., Waltham, MA, USA, C37608 for green, C10045 for orange), Vybrant™ DiO labeling solution (Thermo Fisher Scientific Inc., Waltham, MA, USA, V22886), PKH26 Red Fluorescent Cell Linker Kit (Merck KGaA, Darmstadt, Germany, PKH26GL-1KT), Wheat Germ Agglutinin (WGA) Alexa Fluor 633 (Thermo Fisher Scientific Inc., Waltham, MA, USA, W21404), CellLight™ Plasma Membrane-GFP (Thermo Fisher Scientific Inc., Waltham, MA, USA, C10607), and MemGlow™ 590 (Cytoskeleton Inc., Denver, CO, USA, MG03-02). The cells were labeled with these stains according to the respective manufacturer’s instructions. For each staining method, the dye concentration, or particles per cell depending on the method used, and incubation time were varied to find the optimized labeling results for U87 cells. The best labeling result was then evaluated.

The cells were fixed in a 2% (*w*/*v*) paraformaldehyde (PFA) solution for 15 min at room temperature, washed three times with PBS (Merck KGaA, Darmstadt, Germany, D8537), and finally covered with ProLong™ Gold Antifade Mountant (Thermo Fisher Scientific Inc., Waltham, MA, USA, P36934). The PFA (Merck KGaA, Darmstadt, Germany, 158127) was solved in PBS.

### 2.2. Cell Irradiation

Low-LET X-ray irradiation was performed using a CellRad X-ray irradiation system (Precision X-Ray Inc., Madison, WI, USA) equipped with an integrated dosimeter and a 0.5 mm aluminum filter. The cells were irradiated at a tube current of 5 mA and a tube power of 130 kV, in shelf position 3 of the system, resulting in a dose rate of approximately 1.63 Gy/min. In a previous study, LN229 cells showed a significant reduction in cell survival at 4 Gy [38]; therefore, this dose was chosen to see whether the properties of TNTs were affected by low-LET X-ray irradiation. Irradiation of the sample was automatically stopped when the desired dose of 4 Gy was reached using the automatic dose control of the system. During irradiation, the sample was rotated by the turntable installed in the system to ensure very homogeneous and uniform irradiation of the sample.

### 2.3. TNT Identification

Membrane connections were identified as TNTs when the following criteria were met: two separate cells were connected by the membrane connection, the connection was smaller than 1 µm in diameter, and the connection was located above the substrate. Imaging was performed using only high-quality oil objectives with a numerical aperture of 1.4 to ensure that the resolution was high enough to verify that the diameter was less than 1 µm.

### 2.4. STED and Confocal Microscopy

A Leica TCS SP8 3X STED microscope (Leica Microsystems CMS GmbH, Mannheim, Germany) was used for the cell imaging. The microscope was equipped with a live-cell imaging unit consisting of a climate box and CO_2_ supply with a humidifier (Life Imaging Services GmbH, Basel, Switzerland). Live-cell imaging was performed at 37 °C and 5% CO_2_. The excitation laser used was a White Light Laser (WLL), which is adjustable from 470 nm to 670 nm, allowing for the most appropriate wavelength to be selected for each dye. The selected excitation wavelength and the selected detector range of the hybrid detector (HyD) for each dye are shown in Table 1.

The power of the WLL was kept as low as possible during the live-cell imaging to reduce cell stress. The laser power used for excitation was approximately 1 mW. A scanning speed of 600 Hz was chosen. Additionally, the cells were scanned bidirectionally to reduce motion artifacts and cell stress due to long light exposures. The gating of the signal is shown in Table 1. All samples were acquired as z-stacks.

For STED imaging, a pixel size of 40 nm and a z-step of 160 nm were chosen. Images were captured with a 100× oil objective (Leica HCX PL APO100×/1.4 Oil, Leica Microsystems CMS GmbH, Mannheim, Germany). A STED depletion laser with a wavelength of 592 nm was selected for the CellMask™ Green Plasma Membrane, and a STED depletion laser with a wavelength of 660 nm was selected for the CellMask™ Orange Plasma Membrane. The STED images were acquired with 30% STED laser intensity in the x-y direction and 70% in the z-direction. The STED laser power used was between 50 and 100 mW. The halogen-free and fluorescence-free immersion oil Immersol™ 518 F/37 °C of the company Zeiss (Pulch + Lorenz GmbH, March, Germany, 444970-9010-000) with a refractive index of 1.518 at 37 °C was used as immersion oil.

For confocal imaging, a pixel size of either 40 nm or 80 nm was chosen. The z-step was either 160 nm or 400 nm. Images were acquired using either a 100× oil objective (Leica HCX PL APO100×/1.4 Oil, Leica Microsystems CMS GmbH, Mannheim, Germany) or a 63× oil objective (Leica HCX PL APO63×/1.4 Oil, Leica Microsystems CMS GmbH, Mannheim, Germany). Both objectives had a numerical aperture of 1.4. 

The confocal live-cell videos for the TNT formation analysis were acquired with a pixel size of 80 nm and a z-stack size of 7.6 µm with a z-step of 400 nm. For each sample, a 3 × 3 mosaic image was acquired with a 10% overlap to increase the field size. The 9 images were merged using the “Mosaic Merge” tool of the Leica Application Suite X (LAS X) software (Leica Microsystems CMS GmbH, Mannheim, Germany, version 3.5.6)) into a final image of an approximately 510 µm × 515 µm size for the analysis. The videos had a total length of 6 h, and a time interval of 15 min was chosen. The “Navigator” tool of the LAS X software (Leica) was used to find the same position after the irradiation of the samples using the saved xy-coordinates and with the help of a screenshot of the cells before irradiation. Using the autofocus and a focus map of the imaged field, the cells were kept in focus throughout the imaging period. For the first 2–3 time points, a WLL intensity of 10% was sufficient for a good signal, after which, it was increased to 15%. For the live-cell videos, the cells were labeled with CellMask™ Orange Plasma Membrane Stain. This dye has been shown to give the best signal over a longer period of time, is the most cell-friendly, and labels cells homogeneously. The parameters for the acquisition of the confocal live-cell videos were set so that the cell behavior was not affected by the microscopy itself. Untreated cells proliferated and behaved normally. They did not attempt to escape the imaging field during acquisition.

### 2.5. Image Analysis

To evaluate the labeling methods used, the Signal-to-Noise Ratio (*SNR*) and the mean background value were measured and are presented in Appendix A The *SNR* was calculated as follows:
$$SNR=\frac{{\bar{I}}_{Signal}-{\bar{I}}_{Background}}{{SD}_{Background}}$$


For this purpose, the mean gray value ${\bar{I}}_{Signal}$ of a TNT was measured, as well as the mean gray value of the background ${\bar{I}}_{Background}$ with the standard deviation ${SD}_{Background}$ of the image containing the TNT. For each labeling method, 10–20 *SNR* values were measured, and the mean *SNR* value and the standard deviation of the *SNR* values were calculated. For the evaluation of the background, the mean background values of 10–20 images of each labeling method were measured, and the mean *SNR* value was calculated. All measurements were performed using Fiji.

### 2.6. Deconvolution and SFP Rendering

For the TNT diameter measurements, the acquired images were deconvolved using the Huygens Professional Deconvolution program (version 23.04) from the company Scientific Volume Imaging B.V. (Hilversum, The Netherlands). A theoretical point spread function (PSF) was calculated by the program based on the microscopy parameters used. Deconvolution was performed using an iterative Classic Maximum Likelihood Estimation (CMLE) algorithm. The Simulated Fluorescence Process (SFP) rendering of the TNT in Matrigel was performed with the Huygens SFP Volume Renderer, also available with the Huygens Professional Deconvolution program (Scientific Volume Imaging B.V., Hilversum, The Netherlands, version 23.04). 

### 2.7. Diameter Measuring Using the Autocorrelation Analysis Method of Van Steensel et al.

To measure the TNT diameter, the TNT was excised from the deconvolved STED image, a maximum projection was performed, and the TNT was rotated so that its length was vertically aligned. The TNT diameter was measured using the cross-correlation method introduced by van Steensel et al. [54]. In previous studies, this method has been used several times for the very precise sizing of small objects in STED images [42,55,56]. In this method, the original image and a copy of the original image are shifted to each other pixel by pixel from −Δx to Δx and from −Δy to Δy in the x- and y-directions, respectively, and the Pearson correlation coefficient $r_{P}$ is calculated as follows:$$r_{P}\left(\Delta x,\Delta y\right)=\frac{\sum_{i}\left(A_{i}-\bar{a})(B_{i}-\bar{b}\right)}{\sqrt{\sum_{i}{\left(A_{i}-\bar{a}\right)}^{2}\sum_{i}{{(B}_{i}-\bar{b})}^{2}}}$$
where $A_{i}$ and $B_{i}$ are the gray values at the current pixel *i* in the first image *A* and second image *B*, respectively, and $\bar{a}$ and $\bar{b}$ are the respective mean values. The final result is the 2D cross-correlation function matrix. Since we take a copy of the original image instead of a second image with a different pattern, the autocorrelation function (ACF) is calculated. The ACF is a symmetric correlation problem that can be simplified to a convolution problem where the intensity distribution of the structure is convolved with itself [56]. For simplification, the intensity distribution of a TNT was assumed to be Gaussian. The convolution of two identical Gaussian functions with a width $\sigma_{1}$ again resulted in a Gaussian function with a width $\sigma =\sqrt{2}\sigma_{1}$. For the analysis, the projection of the ACF for the x-shift (Δy=0) was used and a Gaussian function was fitted to the measured ACF. The full width at half maximum (*fwhm*) was determined, and the diameter of the TNT was calculated as follows:$$d_{measured}={fwhm}_{1}=2\sqrt{2\ln2}\sigma_{1}=\frac{fwhm}{\sqrt{2}}$$

Based on previous studies under similar imaging conditions using our setup, a STED resolution of 105 nm was assumed [54]. The actual TNT diameter can be determined by quadratic subtraction of the STED resolution:$$d_{actual}=\sqrt{d_{measured}^{2}-d_{resolution}^{2}}$$

### 2.8. Statistical Analysis

Statistical comparisons were conducted by analysis of variance (Statistic Kingdom, One-Way ANOVA Calculator). The two-sample *t*-test (GraphPad, QuickCalcs, version 9) was used to resolve significant differences, and a *p*-value of ≤0.05 was considered to be statistically significant.

## 3. Results

### 3.1. Suitable Fluorescent Membrane Marker for TNT Research

Since TNT diameters are in the nanometer range, they are only visible under a microscope in cell culture when a good objective with a high numerical aperture (NA) of about 1.4 is used. This can be reached using oil immersion objectives with a magnification of 63× or higher. These objectives have such a high NA and, thus, the necessary resolution. TNTs can be observed in living cells by phase contrast and also by fluorescence microscopy. The advantage of fluorescence microscopy is a better contrast of the TNTs to the background compared to phase-contrast microscopy, and, therefore, fluorescence microscopy allows for better detection of TNTs. However, good labeling of the individual small TNTs is essential for high-quality fluorescence imaging.

Since TNTs are membrane bridges between distant cells, the plasma membrane is the best option to visualize TNTs. In this study, we tried six different membrane dyes to find the most suitable fluorescent TNT marker. Four of these dyes, namely CellMask™ Plasma Membrane, DiO, PKH, and MemGlow™, are lipid inserters. WGA (Wheat germ agglutinin conjugates) labels glycoproteins which are membrane proteins. The final tried dye, CellLight™, is a transduction method. The dyes are evaluated on five critical points: cell-off background noise, the quantum yield in TNTs, the homogeneity of the labeling, the internalization speed of the dye into the cell interior by endocytosis, and the effect on cell viability. For this purpose, the U87 glioblastoma cells were stained according to the respective manufacturer’s protocols and imaged with the Leica TCS SP8 confocal microscope. The results are presented in Table 2. The cell-off background and the quantum yield of the labeling methods were rated as good (+), bad (−), or okay (−/+) depending on the image quality, the WLL intensity used to obtain a sufficient signal, and the overall impression of the staining result of the cells. If the cells were homogenously labeled by the dye used, the homogeneity was rated as good (+), or otherwise, bad (−). To learn more about their function in the survival instinct of the cells, it was necessary to observe the TNTs in living cells for a certain time without the need to repeat the staining to compensate for the signal loss or rearrangement of the dye due to endocytosis. As the dye became more and more rearranged, the signal in the TNTs was becoming lost. Furthermore, due to their small size, TNTs are very sensitive to mechanical stress such as the washing steps in the staining protocol. Therefore, the rate at which the fluorophores bound to the plasma membrane are internalized into the cell body must be as slow as possible, thus maximizing the time interval for qualitative good imaging of the TNTs. To evaluate the suitability of the dye regarding the deposition problem, the residual labeling of the TNTs was checked 1–2 days after the staining. If the signal at the plasma membrane and in the TNTs was still acceptable, the staining was scored as good (+), or otherwise, as bad (−). In the case of CellLight™, which is a transduction labeling method, the internalization speed is not relevant, because all lipids are genetically fluorescent after successful labeling and, therefore, no rearrangement takes place. Whether the staining influences the cell viability, morphology, or behavior depends mainly on the type of loading buffer that must be used to achieve sufficient signal. The best option is to simply use the growth media of the cells, since this generates the smallest amount of cell stress and, for longer incubation times, the cells cannot survive without the usage of growth media. The loading buffers required for the staining methods are also listed in Table 2. Some example images of the U87 cells labeled with these dyes are shown in the Appendix A. Additionally, Appendix A shows the measured Signal-to-Noise Ratios (SNRs) and background values of the analyzed dyes, as well as the WLL intensity used.

The most common problem was that the cells were not stained uniformly. Some cells were very brightly stained, and others were weakly stained or not stained at all. However, if not all cells were sufficiently stained, it was not certain that all TNTs would be detected to analyze the entire TNT network. Only staining with CellMask™ Plasma Membrane and WGA resulted in homogeneous labeling of the U87 cells. All other dyes showed problems here (see Appendix A). Another major problem was a very low SNR considering the signal inside the tubes. In particular, the CellLight™ and WGA dyes showed an SNR below 1, making the detection of the TNTs very difficult. In the case of CellLight™, the overall weak signal produced this poor SNR. Here, even using a high WLL intensity of 85%, which is much too high for long-term live-cell imaging, the transfected cells were difficult to see under the microscope (see Appendix A). An adjustment of the labeling protocol did not result in a higher signal yield, thus, the quantum yield of CellLight™ was rated as bad (−). The bad SNR value of WGA was a result of the overall punctuate staining of the membrane. The vesicles inside the cell were intensely stained, but the plasma membrane and, thus, the TNTs had a punctate appearance. In addition, WGA staining showed the highest average background value (see Appendix A). Therefore, WGA was rated bad (−) in terms of cell-off background and quantum yield. The dye MemGlow™ also showed problems with the cell-off background noise. This dye accumulated strongly on the bottom of the glass and, therefore, generated a lot of cell-off signal (see Appendix A). In addition, the background value was high (see Appendix A). Therefore, the cell-off background noise was rated as bad (−) for this dye. Adjusting the dye concentration or using a serum-free medium did not prevent the accumulation of the dye. The use of normal growth media with serum resulted in less accumulation of the dye, but worsened the inhomogeneous labeling of the cells. The quantum yield for the dyes PKH, DiO, and MemGlow™ was found to be okay (+/−). PKH and DiO yielded SNR values of (1.7 ± 0.8) and (2.5 ± 0.8), respectively. MemGlow™ received an SNR value of (4.8 ± 3.5) in the TNTs when considering well-labeled cells. Since MemGlow™ showed these poor issues regarding dye accumulation and homogeneity problems, the internalization speed was not analyzed for this dye. The internalization of PKH and DiO proceeded fast and, due to the inhomogeneous labeling results, some cells were almost not visible after a few hours (see Figure 1). Furthermore, the PKH labeling protocol was not very good for cell viability due to the use of Diluent C as a loading buffer. 

The best overall results were obtained with the CellMask™ Plasma Membrane dye, which is available as a green and an orange dye. All the cells were uniformly labeled by this dye with an intense and stable signal. The CellMask™ Plasma Membrane obtained the best SNR values with (5.7 ± 3.4) for the orange dye and (5.7 ± 2.9) for the green dye. The background values (11.3 ± 5.7) for the orange dye and (5.7 ± 2.9) for the green dye were also good. Additionally, when the cells were freshly labeled, a WLL intensity of 5% was sufficient for a good signal with low overexpression of the vesicles inside the cells for both green and orange colors. Within an hour, the signal yield decreased, and an increase of the WLL intensity to 10–15% was required for further confocal imaging. After this drop, however, the signal yield remained stable for at least 24 h according to our experience with the orange variant. All other dyes tested required a higher WLL intensity to achieve a sufficient signal. According to the manufacturer, the negatively charged CellMask™ Plasma Membrane dye, regardless of color, is designed to increase the narrowed window for cell membrane studies in living cells by slowing down the internalization process. Indeed, this dye showed a slow internationalization speed, as the plasma membrane of the cells was still sufficiently stained two days after labeling (see Figure 1). In addition, the labeling protocol was very simple, and the cells were labeled within minutes by using the normal growth media as a loading buffer. Therefore, we selected this dye for further live-cell imaging studies in our TNT research. For longer live-cell imaging, the orange variant was preferred, because it is more photostable compared to the green variant. CellMask™ Plasma Membrane labeling also works well for LN229 cells. By cell division, the staining was equally distributed to the daughter cells (see Appendix A). The more cell divisions occurred, the weaker the signal became. However, proper imaging for up to 24 h is possible without the need for fresh labeling.

### 3.2. Size Measurement Using STED Microscopy

A key feature of TNTs is their diameter, which should be in the nanometer range to distinguish them from other membrane compounds, such as epithelial bridges or microtubes. To measure these small sizes, we used STED microscopy with a resolution of about 105 nm. To avoid deformations of the TNTs caused by fixation, we imaged the cells alive. The TNT diameters were measured in the deconvolved images using the auto-correlation method with the van Steensel approach. The advantage of this method is that the diameter size was not measured with the intensity profile at one single position along the structure, but a quantitative diameter profile of the entire length of the structure was used. The results are shown in Figure 2. To eliminate any size bias due to the use of a specific wavelength, the measurements were performed for two types of CellMask™ Plasma Membrane dyes (orange and green) for the U87 cell line. Each measurement was performed once. One sample was used for the measurement with the CellMask™ Orange Plasma Membrane as a dye, and two samples were used for the measurement with the CellMask™ Green Plasma Membrane as a dye. Measurements of the TNT diameter in the U87 cells using the CellMask™ Orange Plasma Membrane dye and CellMask™ Green Plasma Membrane dye resulted in mean values of (194 ± 30) nm for 113 measured TNTs and (197 ± 31) nm for 37 measured TNTs, respectively. The reason for the lower number of TNTs measured with the green variant, even though two samples were used here, was the lower photostability of this dye. The green dye was quickly bleached in the whole sample by the scattered light from the STED laser, making the SNR too poor for an accurate measurement. The ANOVA analysis showed equality of variances and, therefore, allowed for pooling, resulting in a size of (195 ± 31) nm. Since there was no difference in the measured size, the dye causing the least cell stress, CellMask™ Orange Plasma Membrane, was used for the LN229 measurement. The TNTs in LN229 cells had a mean diameter of (338 ± 94) nm, which is significantly larger than the mean diameter measured for the TNTs in the U87 cells (*p* < 0.0001). For the measurement in the LN229 cells, 93 TNTs were evaluated. The LN229 measurement was performed once with three replicates. We measured the TNT diameter at one time and did not acquire live-cell STED videos. Since STED microscopy requires a high laser power to ensure the desired resolution enhancement, the sensitive TNTs and cells might be affected by the imaging itself. Long STED imaging of cells and TNTs often leads to breakage of the sensitive TNTs either by the laser itself or by cell movement as the cells try to escape the imaging field. To prevent the measurement of the TNT diameter from being falsified by the possible influence of the measurement method itself, the measurement, i.e., the microscopy with a strong laser, was kept as short as possible. A measurement by STED video microscopy could result in a falsified smaller TNT diameter because multiple measurements of TNTs that are about to break off would be performed. Therefore, only single images of the TNTs were captured by STED microscopy, not videos. To ensure that the TNT diameter was measured as accurately as possible, a sufficient number of independent TNTs had to be measured at any time in their lifetime. Sufficiently high statistics are particularly important here, since TNTs can adjust their diameter during their lifetime. For this reason, a large number of TNTs were measured in this study. Large bulges such as gondolas and junctions of TNTs were excluded from the measurement. The focus here was kept on the typical TNT diameter, without any special features.

### 3.3. Analysis of Length, Lifetime, and Formation

To find out how TNTs are formed in U87 and LN229 cells, 6 h live-cell confocal microscopy videos were recorded. Confocal microscopy was preferred here because it requires much lower laser intensities compared to STED microscopy, and, therefore, the cell stress is greatly reduced. This costs resolution, but the TNTs were still well visible and the behavior, not the structure, could be well analyzed. The lifetime and formation mechanism of the TNTs were analyzed, as shown in Table 3. For the length measurement, the TNT images acquired in this project were additionally used, resulting in 426 measured TNTs for the U87 cell line and 197 for the LN229 cell line. 

We observed that the TNTs in the U87 and LN229 cells were formed exclusively by cell movement, either after proliferation, when two daughter cells moved apart from each other (see Appendix A), or by cell movement in different morphological forms. A TNT can be formed when two touching cells migrate away from each other (see Figure 3), when a cell pulls an arm back to the cell body that previously touched a neighboring cell (see Appendix A), or when two cells are connected by an EP bridge or microtube and the stretching of this connection due to cell movement generates a TNT (see Appendix A). The TNTs disappear when the cells move together or move too far apart, causing the TNTs to break. In general, TNTs behave like sticky threads between cells that can be stretched to a breaking point. However, they can also become shorter again and, thus, remain stretched rather than bent as the cells move back toward each other again. Their lifetime is probably dependent on their stability and ability to quickly adjust their length. The TNTs in the U87 cells were significantly longer, with a mean length of (44 ± 27) µm compared to those in the LN229 cells, with a mean length of (20 ± 12) µm (*p* < 0.0001). Also, the lifetime of the TNTs in the U87 cells was significantly longer, with (88 ± 60) min compared to the TNTs in the LN229 cells with a mean lifetime of (41 ± 30) min (*p* < 0.0001). Additionally, we observed that the U87 cells had a larger and more complex TNT network than the LN229 cells. The maximum measured length and a recorded lifetime of the TNTs in the U87 and LN229 cells are also shown in Table 3. The differences can be seen more clearly here. The maximum length measured in the U87 cells (189 µm) was three times longer than that measured in the LN229 cells (60 µm), and the maximum recorded lifetime of the TNTs in the U87 cells (360 min) was more than twice as long as that of the TNTs in the LN229 cells (165 min).

In our measurements, we analyzed both untreated cells and cells irradiated with 4 Gy of X-rays prior to imaging. There were no differences in the length and lifetime of the TNTs between the untreated and irradiated cells in either the U87 or LN229 cells (see Appendix A). The type of formation did also not change. The TNTs in the U87 and LN229 cells, whether irradiated or not, were formed exclusively by cell movement. No TNT formation by filopodia growth was found. The ANOVA analysis showed equality of variances; therefore, the measurements of the untreated and irradiated cells were pooled, and the averages of all experiments, as well as the standard deviations, are shown in Table 3. For analyzing the TNT formation in the U87 cells, the cells were imaged for 6 h prior to irradiation. After irradiation with 4 Gy of X-rays, the same position as in the previous video was found again using the “Navigator” tool of the microscope software, and the cells were again imaged for 6 h. This was performed twice independently. For the LN229 cells, the 6 h videos of the untreated cells and the irradiated cells were each performed once independently.

### 3.4. Morphology

Most TNTs appeared as straight thin lines between the cells in 2D cell culture and were usually located above the substrate. In glioblastoma cells, TNT connections can consist of one single TNT (see Figure 4a) or several TNTs (see Figure 4b). In the latter case, it is often difficult to distinguish the individual TNTs from each other because they are tightly packed. TNTs can be found at the full height of the cells (see Figure 4c–e). They can be near the bottom, at the top of the cells, or somewhere between. The four TNTs shown in this image are marked by arrows of different colors. The different heights of the TNTs can be seen nicely in the provided XZ view in Figure 4d and in the simulated fluorescence (Sfp) rendering in Figure 4e. Due to this feature, it is important to image a full z-stack of the cells if one wants to capture all TNTs. TNTs are not necessarily horizontal to the substrate, but are often oblique. One of the most important features of TNTs is their small diameter. This morphological feature is the one that allows them to be distinguished from other larger membrane connections (>1 µm) such as EP bridges or microtubes (see Appendix A). Figure 5 shows some rare morphological features of TNTs in glioblastoma cells. In the U87 cells, we found complex TNT connections linking several cells together by a junctional connection of the TNTs (see Figure 5a,b). A TNT can also be bent over obstacles such as other cells (see Figure 5c). The TNT becomes anchored to the obstacle cell, which is then included in the TNT network and acts as a connected partner (see Appendix A). This feature, as well as the fact that some TNTs branch out at their ends (see Figure 5d), was found in both cell lines. Also, the diameter of a TNT can vary along its length, and bulges or anchored vesicles can be found (see Figure 4d and Figure 6). These bulges may indicate that cargoes larger than the TNT diameter are being transported along the TNTs. In both cell lines, U87 and LN229, large bulges, so-called gondolas, were found. Gondolas can carry enclosed organelles larger than the diameter of the nanotube itself. In Figure 6, the observed gondola is enlarged on the right side to show the contents of the gondola in more detail. We can only speculate what cargo was being transported. However, it can be seen that it carried at least two vesicles and a considerable amount of cytoplasm, which appears completely black due to the absence of lipids. This bulk of cytoplasm could contain several cell organelles, such as mitochondria. Other encapsulated cargoes may be transported within the vesicles. The formation and generation of the force required to move such a bulk vessel are unexplored. The movement may be driven by differences in the chemical potential of the molecules in the bulk solution and the interior of the target cell, or by the composition of the gondola membrane and the target cell membrane [57]. However, we rarely observed such gondolas. In the experiments, only one gondola was found in the 197 connections analyzed in the LN229 cells, and three gondolas were found in the 426 connections analyzed in the U87 cells. Appendix A shows the movement of the gondola shown in Figure 6 along the TNT.

To find out how TNTs appear in a 3D cell culture, we labeled the U87 cells with CellMask™ Orange Plasma Membrane, transferred them into Matrigel, and imaged them directly up to one day after plating. The SNR in deep-tissue-like structures is very low, due to the decreasing laser power. This complicates the analysis of the TNTs in this configuration. A TNT formed in Matrigel is shown in Figure 7. This TNT is no longer straight, but curved, which is consistent with what has been reported in the literature [15,16,58]. Due to their flexibility, TNTs can connect cells even if an obstacle or surrounding extracellular matrix, mimicked here by Matrigel, blocks the shortest path. However, for a kinked or curved morphology, the TNTs might need a supporting material such as other cells, as in Figure 5c, or perhaps Matrigel to provide the necessary stability. An animation video of the SFP rendering shown in Figure 7b can be found in the Appendix A. In this video, the TNT connection can be viewed from different angles. Additionally, the complete acquired z-stack can also be found in the Appendix A. Here, it is easier to see that it is, indeed, a connection compared to the maximum projection shown in Figure 7a. 

### 3.5. Stability during Fixation

The problem of the disruption of TNTs during fixation has been discussed extensively in the literature [23,59]. To find out how the TNT network in the U87 cells was altered by fixation, we performed a total of four experiments. In one experiment, we imaged the fixation process using a phase-contrast microscope (Axio Observer Z1, Carl Zeiss AG, Oberkochen, Germany) with a 63× oil objective (LCI Plan-Neofluar 63×/1.30 Imm Korr Ph 3 M27, Carl Zeiss AG, Oberkochen, Germany) and then acquired confocal images of the fixed cells in one sample. In two further experiments, each with four samples, we first imaged membrane-labeled U87 cells alive, then we fixed the cells and acquired further images of the fixed cells from the same region as previously imaged alive with the help of the navigator software and using dishes with a grid. Each experiment resulted in an enormous loss of the fine membrane structure. Some images and the phase-contrast video acquired during the fixation process are shown in the Appendix A.

To find out at which point the TNTs break during the fixation, in the fourth experiment, we imaged membrane-labeled U87 cells before, during, and after fixation without removing the sample from the microscope. In this experiment, the cells were first imaged alive under live-cell conditions, the medium was carefully removed, and 2% PFA (in PBS) was added dropwise while the system was allowed to cool down to room temperature. Immediately after the addition of the 2% PFA solution, we started to acquire a video of the cells for the incubation time of 15 min (see Appendix A). After fixation, we carefully washed the cells three times with PBS and imaged the sample again. Finally, we carefully dropped the ProLong Gold mounting medium on the sample and acquired images during the curing time of 26 h after fixation. Since the sample was never removed from the microscope, the same area of the sample was imaged for all conditions, allowing for a direct identification of the breakage. Figure 8 shows the maximum projections of the confocal microscopy images of the steps (alive, immediately after fixation, after 3× PBS wash, and after 26 h of ProLong Gold cure time). It can be seen that the cells moved very little between the live-cell imaging (Figure 8a) and the post-fixation imaging (Figure 8b). The TNTs were not destroyed by the fixation process itself. We did not see any TNT breakage on the video, but the TNTs did tremble. This trembling was probably due to thermal fluctuations, as these dynamics are visible in live-cell videos when the temperature changes. Nevertheless, it seems that the structures lost their flexibility after the fixation process and washing after fixation made the structures disappear (see Figure 8c). The situation became even worse when the ProLong Gold mounting medium was added (see Figure 8d). Almost all small membrane structures were lost here. Additionally, artifacts appeared (see orange circle in Figure 8d). These artifacts were probably caused by lipid interactions with the ProLong Gold. The artifacts and loss of the fine membrane structures occurred immediately when ProLong Gold was put on the sample. Overall, it seemed that, due to the loss of flexibility and, therefore, the loss of ability to survive stretching, compression, and bending, the TNTs and other fine membrane structures became very sensitive stiff structures that were easier to break and, thus, were easily washed away by further steps after the fixation itself.

## 4. Discussion

In our studies, we found that the CellMask™ Plasma Membrane stain is the most suitable dye for labeling the membrane of glioblastoma cells and, therefore, for visualizing TNTs. It gives an intense signal that is sufficient for STED microscopy, and cells are labeled homogeneously. The rearrangement of this dye by endocytosis is slow enough to allow for live-cell studies for several hours. In our project, we performed live-cell studies where we imaged cells for 24 h without the need for additional labeling of the cells. Thus, this dye is a good option for TNT research, which is also consistent with other reports [59]. We measured a mean TNT diameter of (197 ± 31) nm in the U87 cells. The result is below the resolution limit of a confocal microscope, indicating that higher-resolution microscopy methods, such as STED microscopy, are needed for accurate measurement of TNTs. The TNTs in the LN229 cells were significantly larger, with a mean diameter of (338 ± 94) nm (*p* < 0.0001). Both mean diameters measured were below 1 µm, demonstrating that these structures were nanotubes. This also shows that high-quality objectives with good NA are needed to ensure that the resolution is high enough to see these fine structures and not just larger membrane bridges. It is also important to acquire z-stacks of the full height of the cells to detect all TNTs, as TNTs can be located at different heights (see Figure 4). In our laboratory, we established the acquisition of long-term live-cell videos of membrane-labeled glioblastoma cells using confocal microscopy. Instead of confocal microscopy, one could also imagine using holotomography for live-cell imaging studies. This relatively new technology measures the physical properties of a cell, i.e., the refractive index. The major advantage of this method is that the cell membrane can be studied without the need for a fluorescent marker. This avoids problems such as phototoxicity, photobleaching, or interference of the dye with cell dynamics. In addition, the lateral resolution of holotomography is less than 100 nm, allowing for high-resolution microscopy of living cells [60]. Holotomography image acquisition is faster and requires a lower laser intensity compared to fluorescence confocal microscopy. This reduces stress on living cells. The use of holotomography as a microscopy method would also solve the problem of the limited imaging window caused by the internalization of the membrane dye.

In the U87 cells, we measured a maximum TNT length of 189 µm, and in the LN229 cells, a maximum length of 60 µm. The mean TNT lengths of (20 ± 12) µm and (40 ± 30) µm were measured for TNTs in the LN229 cells and U87 cells, respectively. These results again show that direct cellular communication via TNTs can cover a long distance and is, therefore, well-suited for glioblastoma cells to exchange information rapidly. The TNTs in the U87 cells appeared to be more stable than those in the LN229 cells. Although they were thinner, they had a significantly longer lifetime and were also significantly longer (*p* < 0.0001). The better stability of the TNTs in the U87 cells could also explain why more TNT junctions and complicated cross-linking patterns were observed in the U87 cells. To prove whether TNTs in U87 cells are more stable than those in LN229 cells, it is necessary to perform elastic measurements, for example, with optical tweezers. Such measurements have already been performed with U87 cells, but to our knowledge, not with LN229 cells [49]. We believe that the stability of the TNTs depends on their ability to expand and shrink in a very dynamic pattern. This requires that the cells provide a sufficient lipid and cytoskeletal reservoir to supply the TNTs with the necessary assemblies. U87 cells have a mean size of (1900 ± 1400) µm^2^, and LN229 cells have a mean size of (2000 ± 1300) µm^2^. This means that U87 cells and LN229 cells are the same size. However, U87 cells are much more elongated and tend to form long membrane extensions compared to LN229 cells, which have a more compact morphology. This can be seen by looking at the perimeter of the cells. With a mean perimeter of (320 ± 119) µm, U87 cells have a significantly larger perimeter than LN229 cells with a mean perimeter of (205 ± 58) µm (*p* < 0.0001). One can assume that U87 cells need more cell membrane components and also more cytoskeleton to take on these elongated shapes. Therefore, this morphological feature of U87 cells may indicate a larger lipid and cytoskeletal reservoir for TNT formation and stability. Cell size and perimeter measurements were performed by measuring the cell area using the “freehand selections” tool of Fiji to mark the cell boundary. In total, 100 cells were measured per cell line. Phase-contrast images of the cultured cells were used for the cell size and perimeter measurements.

In this study, the focus was on glioblastoma cell lines. However, TNTs from other cell lines such as astrocytes or fibroblasts with different lipid concentrations should be analyzed to answer the question of whether the lipid content is critical for TNT stability and morphology. Here, fluorescence imaging might be challenging if the lipid content is too low, and, therefore, the fluorescence signal of the membrane becomes weak. Nevertheless, one could think about performing such studies using label-free imaging methods such as holotomography, which provides a good resolution and is well-suited for live cell imaging. We would also like to mention that it is also important to study TNTs between different cell types. In the tumor microenvironment, cancer cells are mixed with a variety of other cell types, such as immune cells and normal cells. For accurate cancer research, it is important to find out how cellular communication along TNTs might work in such complex environments and perhaps differ from the isolated cultivation of cancer cells or other cells alone. Here, co-culture studies would be an option to shed more light on TNT communication between different types of cells.

Table 4 provides a comparison of the TNT diameters, lengths, and lifetimes of the two glioblastoma cell lines studied, U87 MG and LN229, and other cell lines found in the literature. Similar to the glioblastoma cell line LN229, Jurkat T cells and HeLa cells have TNTs with a mean length of about 20 µm. TNTs in natural killer cells and the human retinal pigment epithelial cell line ARPE-19 have mean lengths of approximately 30 µm and 44 µm, respectively. This is similar to the TNTs of the U87 MG cells with a length of (40 ± 27) µm. The reported maximum TNT lengths for Jurkat T cells, HeLa cells, natural killer cells, and ARPE-19 cells are 100 µm, 40 µm, 140 µm, and 120 µm, respectively. Thus, we measured a higher maximum TNT length in the U87 cells with 189 µm. The maximum TNT length of 60 µm for LN229 cells is somewhere in the middle range. The lifetimes of the TNTs reported in the literature range from a few minutes to less than an hour and up to several hours. In addition, the diameters we measured for the TNTs in the U87 and LN229 cells are similar to the diameters reported for PC12 cells, Jurkat T cells, and ARPE-19 cells.

Besides the components inside the TNTs themselves, the TNTs can become kinked or bent if other objects are present to provide additional stability. This happens when a third cell supports the connection, or when the cells are cultured in Matrigel (see Figure 5 and Figure 7). Otherwise, the TNTs appear as straight structures. Unfortunately, imaging TNTs in 3D culture is very difficult due to the poor SNR caused by the blocking of the laser and scattering from the Matrigel. For better imaging, methods other than confocal microscopy may be more appropriate, such as two-photon excitation microscopy. 

TNTs in U87 and LN229 cells are formed exclusively by cell movement as two cells separate. TNT formation by filopodia growth, i.e., actin polymerization, was not observed by us in glioblastoma cells. However, we cannot exclude the possibility that TNT formation by filopodia growth occurs in glioblastoma cells. It may be that this type of TNT formation is very rare and, therefore, difficult to observe. It could also be that TNT formation by filopodia growth is only activated under certain conditions with specific triggers. We could see no difference between the formation of a single TNT and the formation of a complex connection consisting of several TNTs. Both types of connection, simple and complex, are formed by cell movement. Since cell movement seems to be the only mechanism by which TNTs are formed, it can be assumed that the migratory behavior of the cells plays a crucial role in TNT communication, and perhaps vice versa. TNTs might play a role in the guidance of cells. It has been reported that TNTs are often found at the migration front [61]. We also observed this behavior. We imaged migrating U87 cells labeled with CellMask™ Orange Plasma Membrane. Here, we observed TNT formation between the cells at the front and the cells behind when the cells moved apart (see Appendix A). TNTs may act as a communication tool for feedback loops, telling the other cells to follow or stay in place. However, this function has yet to be proven and further live-cell studies are needed to shed more light on this hypothesis. Furthermore, in our confocal videos, we could not address whether the TNTs were open-ended or closed-ended, and whether the lumen of both connected cells was contained in the TNTs or not. To answer these questions, studies with higher resolution, as well as with the different labeling of certain substances present in the cell interior, are needed. Here, co-culture studies of cells with differently labeled cytoplasm could provide information on whether there is a one-way communication or two-way communication along TNTs. 

In our studies, we found that neither the lifetime nor the length and formation type of TNTs were affected by low-LET X-ray exposure. Whether the TNT frequency or the linkage rate of the TNT network is affected by this type of radiation remains to be investigated. For these studies, a higher number of repetitions is necessary to be able to make statistically significant statements. Here, the small number of repetitions was sufficient, as many individual TNTs were measured and, thus, resulted in high enough statistics. When evaluating the percentage of connected cells or the connection rate, only 1–2 values per time point could be measured in these experiments. These values were not meaningful enough. For this reason, this analysis was not carried out here. It has been reported that radiation affects the migratory behavior of glioblastoma cells, which may be dependent on the type of radiation [38,39]. We believe that it is important to discover the role of cellular communication via TNTs in increased radiation-induced migration and infiltration rates. It would be beneficial to be able to inhibit TNT formation. Cytochalasin B has been shown to reduce the number of TNTs [62,63,64,65]. However, this toxin stops the polymerization of actin filaments and, thus, has a strong effect on the cytoskeleton of the cells. The cytoskeleton is, without any doubt, necessary for the cell to maintain its shape and move forward. Therefore, we do not think that cytochalasin B is suitable for investigating the role of cellular communication via TNTs in the migration behavior of cells. We need to find another TNT inhibitor for glioblastoma.

In this context, it is also important to find out what kind of cargo is transported by the TNTs. We found that large carriers, called gondolas, are rarely present in glioblastoma cells. Therefore, we think that this is not the dominant communication mechanism of TNTs in glioblastoma cells. As a possible mechanism of therapy resistance, the transfer of calcium through membrane tubes has been reported in astrocytoma cells treated with X-rays [40]. In addition, there are several other cargoes, including mitochondria, microRNAs, and p-glycoprotein, that can promote resistance to therapy [10,13,66,67]. These are all candidates for further TNT research.

In our studies, we found that the formaldehyde fixation process itself does not destroy TNTs. However, the fixation probably takes away their flexibility so that they can no longer adapt elastically to their environment and, thus, break off more easily. TNTs are mostly destroyed during further sample preparation steps such as washing or mounting. As a conclusion, we encourage other researchers to prefer live-cell studies to learn more about the functions of TNT networks in cells. We did not investigate other fixatives, such as glutaraldehyde. It has been reported by others that glutaraldehyde is more suitable for TNT studies with fixed cells [23,59,68]. Glutaraldehyde fixation results in a higher cross-linking rate than PFA fixation [69]. For this reason, glutaraldehyde fixation often provides the best preservation of cellular ultrastructure [70] and is, therefore, well = suited for electron microscopy [69]. However, glutaraldehyde fixation generates free aldehyde groups that severely affect immunohistochemistry results with high background levels [69]. The free aldehyde groups must be blocked or quenched before attempting immunohistochemical staining [70]. As a result, glutaraldehyde fixation is not usually used for immunostaining. Instead, PFA fixation is the most commonly used fixation method for immunostaining [70,71]. Many proteins and cellular organelles can be labeled by fluorescent immunostaining. Therefore, immunostaining is often used to study potential cargos or components inside TNTs [24,65,72,73]. This motivated us to analyze TNT breakage during sample preparation in a commonly performed PFA fixation protocol. However, it is also important to find out how other fixation protocols using glutaraldehyde or a mixture of PFA and glutaraldehyde affect TNT networks. It could be that stronger cross-linking might provide a better stability against shear forces, and this could be the reason why some researchers have found that glutaraldehyde fixation is more suitable for TNTs. Nevertheless, it has been reported that TNTs are often lost due to cell fixation, regardless of the fixative used [7,68,72,74]. In our opinion, there are many possibilities to label organelles or other cellular components in living cells, thus reducing the need for fixation in TNT research.

However, for some applications, such as scanning electron microscopy or cryo-electron microscopy, it is still necessary to fix the cells. These methods are important for TNT research, in particular for revealing the essential structural features of TNTs [23,74]. We would like to point out here that researchers should be aware of the loss of TNTs due to sample preparation. Fixation should only be used if the application requires it.

**Table 4 cells-13-00464-t004:** Comparison of TNT properties among different cell types. N.A. means not announced.

Cell Line	Diameter [nm]	Mean Length [µm]	Maximum Length [µm]	Lifetime	Reference
U87 MG	195 ± 31	40 ± 27	189	Mean (88 ± 60) min; max 360 min	Studied here
LN229	338 ± 91	20 ± 12	60	Mean (41 ± 30) min; max 165 min	Studied here
PC12	50–200	15–60	N.A.	<250 min	[7,62]
Jurkat T cells	180–380	22 ± 3	>100	<1 h	[68]
ARPE-19	50–300	43.6 ± 18.1	>120	<1 h	[75]
HeLa	N.A.	17.7 ± 8.3	40	N.A.	[76]
Human monocyte-derived macrophages	Approx. 700	N.A.	N.A.	Several hours	[77]
Normal rat kidney cells	N.A.	10–70	70	Several hours	[24]
Natural killer cells	N.A.	Approx. 30	<140	Observed for 40 min	[20,78]

## 5. Conclusions

In conclusion, we characterized TNTs in two glioblastoma cell lines, U87 and LN229. We investigated their basic geometry, morphological features, lifetime, and formation. Furthermore, we found out which membrane labeling method is most suitable for TNT research and showed that sample preparation causes the breakage of many TNTs. As a result, we suggest that live-cell studies should be used whenever possible to learn more about the functions and roles of TNTs. We believe that TNTs play an essential role in the migratory behavior of cells as they are formed by cell movement. We hypothesize that TNTs may help cells to orient themselves and possibly transmit feedback. To prove this hypothesis and to learn more about the role of TNTs in the development of therapy resistance, more research on TNTs is needed. Since this type of cellular communication gives cells the ability to exchange many kinds of cargo in a very efficient way, it is reasonable to assume that TNTs play a crucial role in cell survival. By learning more about TNTs and understanding their role in cancer, we may find new ways to treat cancer by using our knowledge of how cells interact with each other.

## Figures and Tables

**Figure 1 cells-13-00464-f001:**
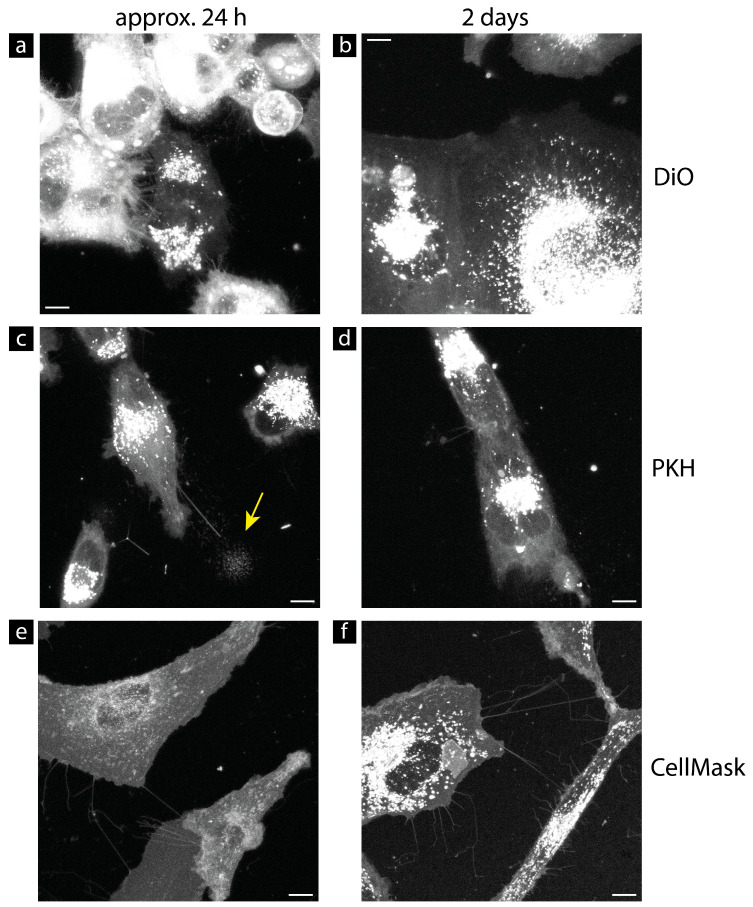
Remaining staining results one day and two days after labeling of DiO (**a**,**b**), PKH (**c**,**d**), and CellMask™ Plasma Membrane (**e**,**f**) in U87 cells. The main problem with DiO and PKH is the inhomogeneous labeling. After 24 h, there are some cells labeled with DiO whose plasma membrane is barely visible (**a**). The same problem occurs with PKH. After 21 h, some cells are hardly visible, and only some vesicles are visible ((**c**), yellow arrow). The internalization of the dye increases the signal of the vesicles inside the cells, so that after 2 days, the vesicles are completely overexposed, while the plasma membrane shows only a very weak signal (**b**,**d**). The internalization of the CellMask™ Plasma Membrane stains is slower compared to DiO and PKH. (**e**) At 26 h after the labeling with CellMask™ Green Plasma Membrane. (**f**) At 2 days after the labeling with CellMask™ Orange Plasma Membrane, the membrane as well as small membrane structures such as filopodia and TNTs are still well visible, and also the SNR is still good. All images are maximum projections of confocal images. Brightness and contrast were not adjusted. Scale bar: 10 µm.

**Figure 2 cells-13-00464-f002:**
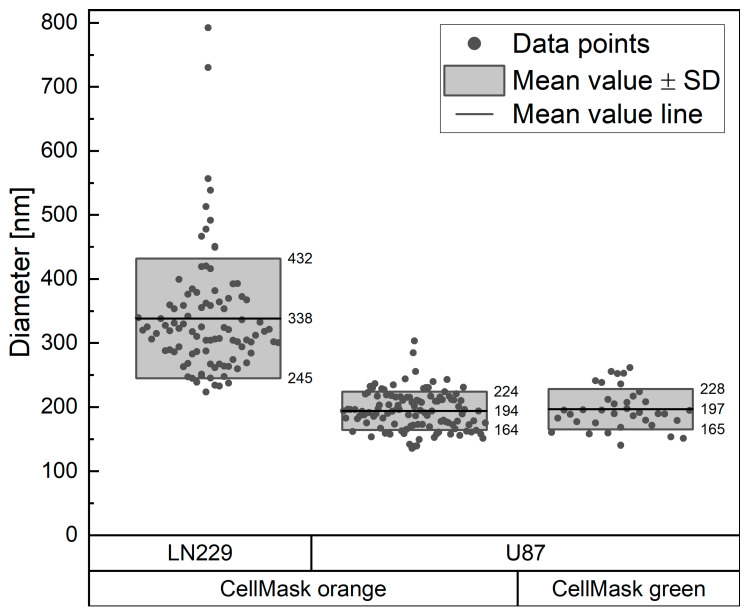
TNT diameters of LN229 and U87 cells were measured by STED microscopy. One point represents one TNT. The lines show the mean values and the borders of the boxes show the range within the standard deviation (SD). Measurements included 93 TNTs from LN229 cells, 113 TNTs from U87 cells labeled with CellMask™ Orange Plasma Membrane, and 37 TNTs from U87 cells labeled with CellMask™ Green Plasma Membrane.

**Figure 3 cells-13-00464-f003:**
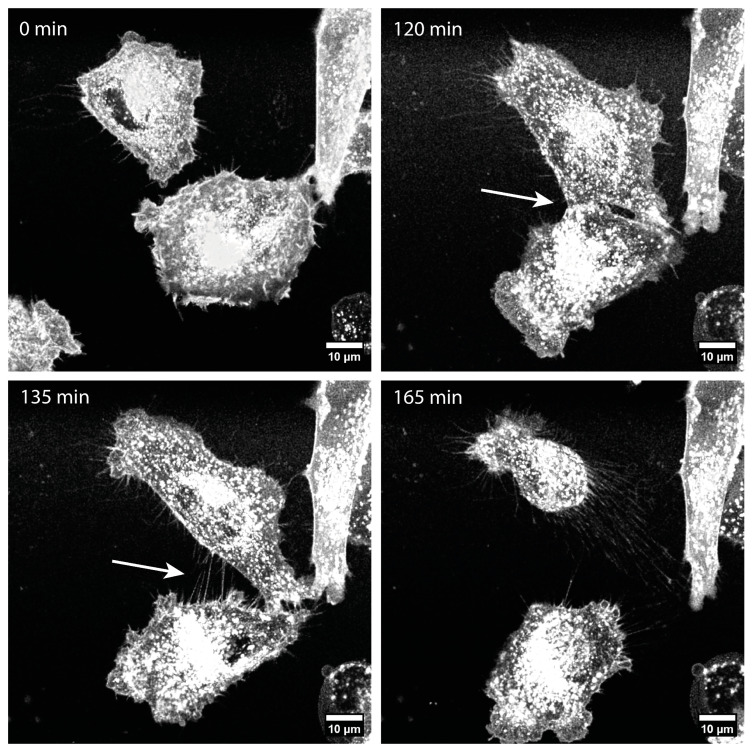
TNT formation by cell movement in LN229 cells labeled with CellMask™ Orange Plasma Membrane. Images are maximum projections of confocal images. The white arrow marks the position of TNT formation at 120 min and the formed TNTs at 135 min. At 165 min, the TNTs have already been torn off.

**Figure 4 cells-13-00464-f004:**
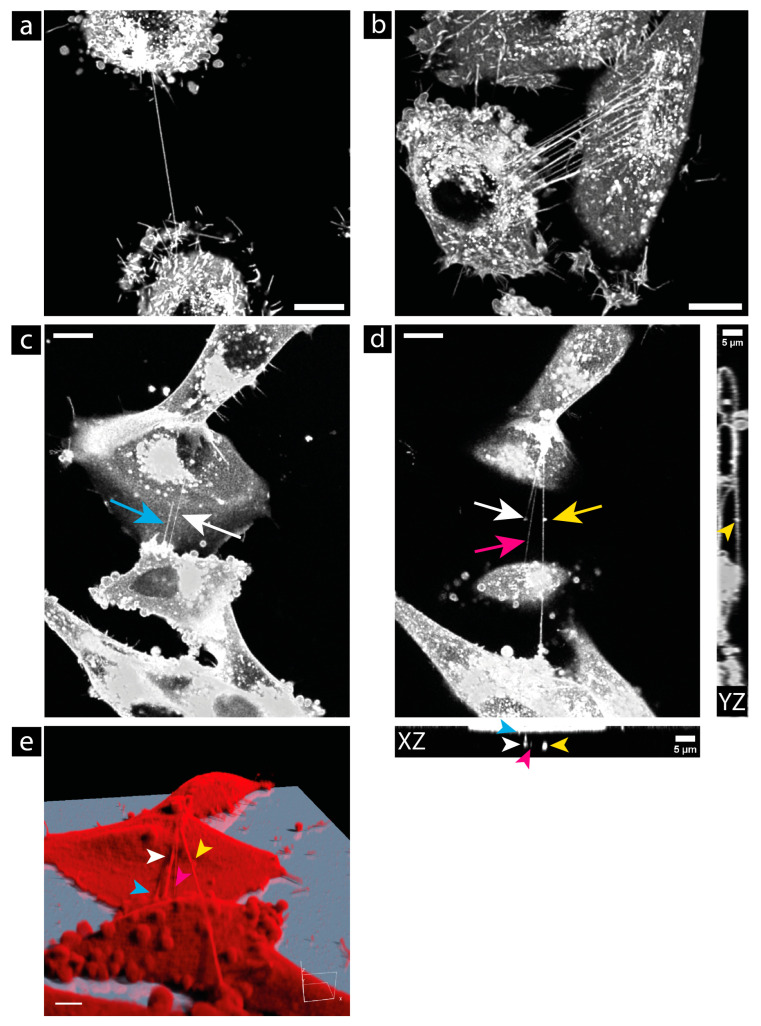
Glioblastoma cells can be connected by a single TNT as a simple connection (**a**) or by several TNTs as a complex connection (**b**). TNTs can be found at different heights in the cells, closer to the bottom (**c**) or even at the top of the cells (**d**). TNTs may have bulges along or anchored to the tube visible in ((**d**), tip of yellow arrow). In (**d**) the orthogonal views YZ and XZ of the TNT connections are shown. (**e**) Simulated Fluorescence Process (SFP) volume rendering of the entire acquired z-stack of the image partially shown in (**c**,**d**). The four TNTs visible in (**c**–**e**) are marked by arrows of different colors (white, yellow, blue and magenta). (**a**,**b**) Show membrane-labeled LN229 cells. (**c**,**d**) Show membrane-labeled U87 cells. CellMask™ Orange Plasma Membrane was used for the staining. All images are maximum projections of confocal images. Scale bar: 10 µm and 5 µm in orthogonal views.

**Figure 5 cells-13-00464-f005:**
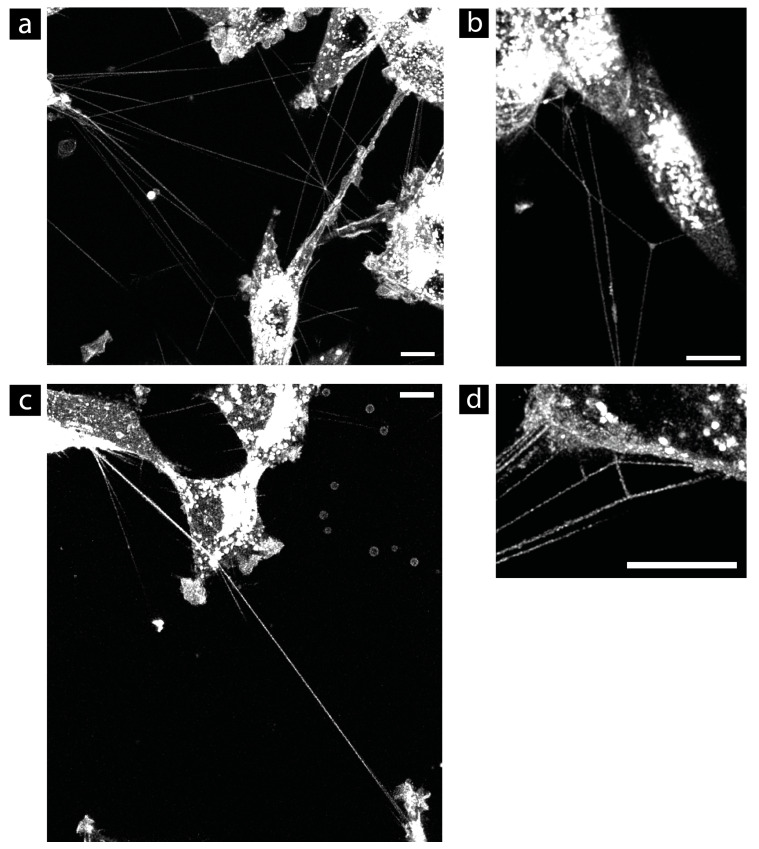
Morphologic features of TNTs in glioblastoma cells. (**a**) In U87 cells, star-shaped connections can be found, where many cells are connected by a junction. (**b**) Triple connection by a junction. TNTs can also be bent over another cell (**c**) or can have branches (**d**). All cells shown are U87 cells labeled with CellMask™ Orange Plasma Membrane. All images are maximum projections of confocal images. Scale bar: 10 µm.

**Figure 6 cells-13-00464-f006:**
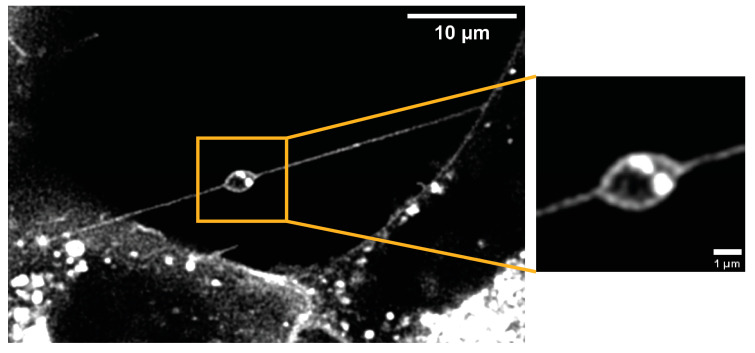
Transport of cargo within a large bulge, called a gondola, along a TNT connection in U87 cells labeled with CellMask™ Orange Plasma Membrane. Images are maximum projections.

**Figure 7 cells-13-00464-f007:**
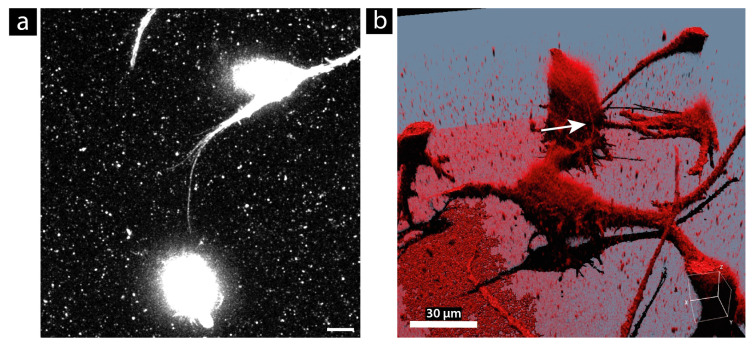
TNT connection of membrane-labeled U87 cells cultured in Matrigel. (**a**) Maximum projection of the top acquired slices of the z-stack. (**b**) Simulated Fluorescence Process (SFP) volume rendering of the entire acquired z-stack. The white arrow marks the TNT. Cells were labeled with CellMask™ Orange Plasma Membrane. Scale bar: 10 µm (**a**) and 30 µm (**b**).

**Figure 8 cells-13-00464-f008:**
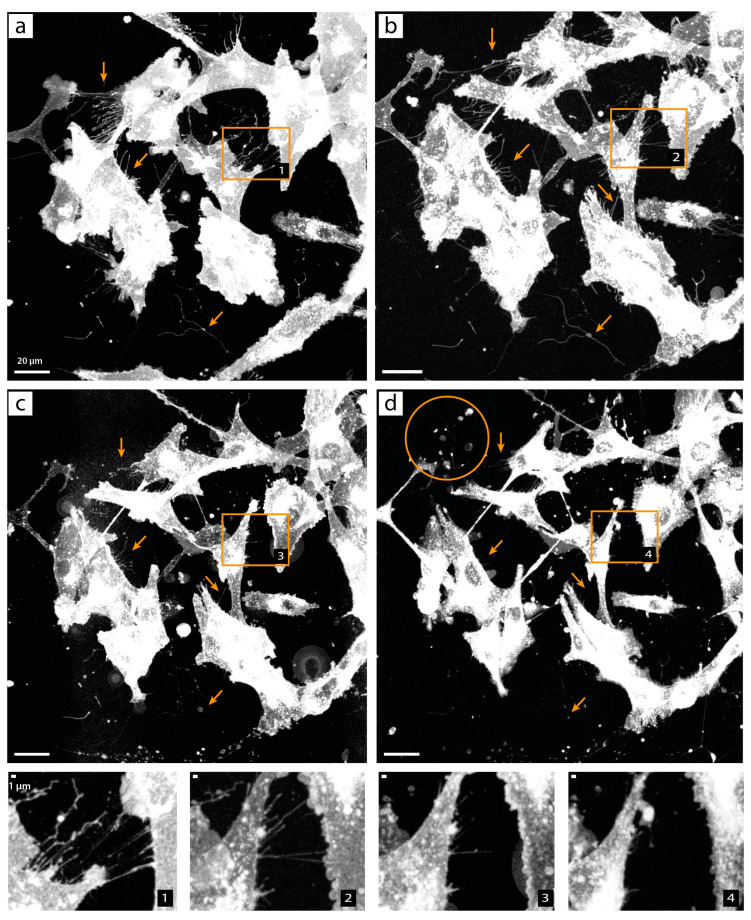
Breakage of TNTs and filopodia in U87 cells during fixation protocol. U87 cells labeled with CellMask™ Orange Plasma Membrane Stain are imaged (**a**) alive, (**b**) 17 min after the addition of 2% PFA solution, (**c**) after fixation and three washes with PBS in PBS, and (**d**) fixed in ProLong Gold after 26 h curing time. An enlargement of each condition is shown at the bottom. The number in the lower right corner of the enlargements indicates the respective section. Arrows indicate structures of interest that were lost during the fixation protocol. The orange circle in (**d**) marks artifacts caused by lipid interactions with ProLong Gold. Scale bar: 20 µm in the overview, 1 µm in the enlargements.

**Table 1 cells-13-00464-t001:** Excitation wavelengths and detector ranges used for all dyes.

Dye	Excitation Wavelength [nm]	Detector Range [nm]	Gating [ns]
CellMask™ Orange Plasma Membrane Stain	554	567–635	0.3–6
CellMask™ Green Plasma Membrane Stain	522	530–583	0.3–6
Vybrant™ DiO	484	501–583	0.3–7
PKH	551	559–630	0.3–12
WGA Alexa Fluor 633	625	634–751	0.3–6
CellLight™ Plasma Membrane-GFP	485	500–580	0.3–6
MemGlow™	595	613–750	0.3–6

**Table 2 cells-13-00464-t002:** Evaluation of the five membrane stains used. (+), (−/+), and (−) mean good, okay, and bad, respectively. N.A. indicates not analyzed.

Dye	Cell-Off Background Noise	Quantum Yield	Homogeneity	Internalization Speed	Loading Buffer
CellMask™ Plasma Membrane	+	+	+	+	Medium
DiO	−/+	−/+	−	−	Medium
PKH	−/+	−/+	−	−	Diluent C (iso-osmotic, aqueous solution)
WGA	−	−	+	N.A.	HBSS
CellLight™	−/+	−	−	Not relevant	Medium
MemGlow™	−	−/+	−	N.A.	Serum-free Medium

**Table 3 cells-13-00464-t003:** Length, lifetime, and formation type of TNTs in U87 MG and LN229 glioblastoma cells.

Cell Line	Mean Length ± SD [µm]	Max. Measured Length [µm]	Mean Lifetime ± SD [min]	Max. Recorded Lifetime [min]	Formation Type
U87 MG	40 ± 27 *n* = 426	189	88 ± 60 *n* = 234	360	Cell movement
LN229	20 ± 12 *n* = 197	60	41 ± 30 *n* = 85	165	Cell movement

## Data Availability

The data presented in this study are available on request from the corresponding author. The data are not publicly available due to restrictions set by the university.

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
