# Peer review of "High-Resolution Microscopic Characterization of Tunneling Nanotubes in Living U87 MG and LN229 Glioblastoma Cells"

_cells, 2024, doi:10.3390/cells13050464_

Round 1

Reviewer 1 Report

Comments and Suggestions for Authors

The authors studied tunneling nanotubes (TNTs) in glioblastoma cancer cells. Glioblastoma poses treatment challenges due to therapy resistance and aggressive features. The study investigates TNT properties in two glioblastoma cell lines, U87 MG and LN229, including diameter measurement, length, morphology, lifetime, and formation using high-resolution live-cell microscopy. It suggests that understanding cellular organization via TNT communication may lead to new therapeutic approaches. Membrane-labeling methods suitable for studying TNTs are discussed, advocating for live-cell studies to explore TNTs' role in cellular behavior. Observations indicate TNTs' potential involvement in cell migration and guidance.

The study is well written and suitable for publication after addressing these minor comments:

The authors should discuss the formation of the TNTs more. Are they formed by two cells separating or are they formed by the polymerization of actin that pushes out from the cell? Do the TNTs include the lumen that connects both cells? Are they open ended or close eneded? The proof of formation of TNTs should be stressed more. Here, a reference to TNT formation should be made in the introduction at the appropriate place: "https://doi.org/10.3390/cells8060626".

The authors could compare the investigated TNTs with other similar TNTs found in the literature. Namely the length, width and formation could be compared. 

The authors claim that only one such gondola was observed. What do they presume the gondola is carrying?

Other problems:

line 154: "Error! Reference source not found.. "
line 160: "n Error! Reference source not found.. "

line 353, line 262, 285, 378, 384... "Error! Reference source not found.."

Comments on the Quality of English Language

The Discussion section could use some upgrade to its readability by breaking the long paragraphs into shorter, more concise ideas. Otherwise the English is fine throughout. 

Reviewer 2 Report

Comments and Suggestions for Authors

Reviewer 3 Report

Comments and Suggestions for Authors

The subject of this manuscript is interesting, but the formatting seems to have serious problems: the same images appear many times, there are problems with the citation of references, and furthermore, I have been unable to obtain access to the supporting information, so I regret that I cannot give a thorough critique of this manuscript and hope to review it again after the issues I have raised have been addressed.

Author Response

We thank the reviewer for the comments. 

We apologize for these circumstances. Something must have gone wrong because the version of the manuscript we uploaded was fine with the figures and references. However, we have corrected the missing figure names as well as table names and deleted the duplicate figures. If we made a mistake in uploading or releasing the supplementary material. Please let us know.

Round 2

Reviewer 2 Report

Comments and Suggestions for Authors

In general, I would advise authors to improve the aesthetic quality of their figures and supplementary materials, thus improving the overall presentation and impact of their work.

Author Response

We thank the reviewer for this comment. We have revised the layout of the figures in both the manuscript and the Supplementary Material. We have changed the arrangements to make them look more beautiful and improve their overall impression.

Reviewer 3 Report

Comments and Suggestions for Authors

First of all, I am sorry that due to the shortcomings of the previous version and the lack of SI authority, I was unable to provide review comments in time. This may cause the author to extend the publication time. Please understand.

This manuscript tells a story about how the author observed TNT in gliomas and obtained TNT-related parameters. The content of the story is relatively novel and interesting. The relevant parameters obtained by the author are relatively rich and the data is very detailed. It is a work worth publishing, but the following questions should be responded to or resolved before publishing

Major:

1.     Why only three dyes are showed in Figure 1 since the authors used and studied 6 dyes in total?

2.     Line 382 to 385, the authors mentioned “One sample was used for the measurement with the CellMask™ Orange Plasma Membrane as a dye, and two samples were used for the measurement with the CellMask™ Green Plasma Membrane as a dye”, but in figure 2, 113 CellMask™ Orange-stained TNT is counted while only 37 for CellMask™ Green, seems opposite, so please make sure the statement is correct.

3.     In figure 3, no obvious TNT can be observed where the arrow points out in the right up picture.

4.     Line 648 to 650, the authors mentioned “Cell size and perimeter measurements were performed by measuring the cell area using the “freehand selections” tool of Fiji to mark the cell boundary. 100 cells were measured per cell line.”, I’d really recommend the authors use the “Threshold” tool to achieve this to increase the efficiency and accuracy. Here’s resource for your reference: https://imagej.net/ij/docs/guide/146-28.html#sub:Threshold...[T]

Minor:

1.     The full name should be given when first time appear of abbreviation like STED,

2.     Some format mistake in line 162, 169, 456, 471, 474, 490, 517, 521, 569

3.     Line 134, the fixation temperature should be mentioned.

4.     Line 276, “(+/-)” should be “(-/+)”.

5.     Line 590, delete “in the solution”.

6.     Some videos in SI cannot be opened by normal software like video 1, 5, and 11, please optimize it if possible.

7.     The layout and arrangement of the pictures are not reasonable and beautiful. For example, in Figure 8, the enlarged images of 1, 2 and 3, 4 should be placed on the right side of a, b and c, d respectively.
